# Predicting Public Adherence to COVID-19 Preventive Measures: A Cross-Sectional Study in Hong Kong

**DOI:** 10.3390/ijerph182312403

**Published:** 2021-11-25

**Authors:** Elsie Yan, Daniel W. L. Lai, Vincent W. P. Lee, Haze K. L. Ng

**Affiliations:** 1Department of Applied Social Sciences, The Hong Kong Polytechnic University, Hong Kong, China; vwp.lee@polyu.edu.hk (V.W.P.L.); haze-kl.ng@polyu.edu.hk (H.K.L.N.); 2Faculty of Social Sciences, The Hong Kong Baptist University, Hong Kong, China; daniel_lai@hkbu.edu.hk

**Keywords:** COVID-19, pandemics, preventive measures, adherence, health belief

## Abstract

**Objectives:** To effectively control the spread of COVID-19, the public’s adherence to relevant disease preventive measures (DPM) is critical. This study examined individuals’ adherence to various DPM and identified facilitators and barriers to adherence in a community sample in Hong Kong. **Methods:** In this cross-sectional study, telephone surveys were conducted over December 2020 and January 2021. Participants provided responses on their adherence to DPM as well as other psychosocial and cognitive factors via the phone. **Results:** Of the sample of 1255 Chinese adults (aged >18 years, 53% women), 94.4% wore face masks in public areas; 88.4% avoided touching their eyes, nose, and mouth; 82.1% performed hand hygiene practices; 81.5% used alcohol-based hand rubs; 74.6% abided by social distancing; and 39.7% tested for COVID-19 on a voluntary basis. Perceived benefits, perceived barriers, self-efficacy, cues to action, perceived acceptability, and disruptions to daily life related to COVID-19 were associated with individuals’ adherence to DPM. **Conclusions:** Adherence to DPM was strong in Hong Kong, and the adherence level could be predicted by various factors. It is vital to consider these factors in order to improve the public’s adherence.

## 1. Introduction

Since the first cluster of cases was reported in Wuhan, China, in late 2019, the Coronavirus disease 2019, commonly known as COVID-19, has brought substantial challenges to public health throughout the world with its high severity and infectivity. In response to this alarming global health crisis, the World Health Organization (WHO) announced COVID-19 as pandemic in March 2020. As of 10 June 2021, there have been over 174 million confirmed cases and almost 4 million deaths reported by the WHO [1].

In the absence of definitive pharmaceutical solutions to combat COVID-19, self-imposed disease preventive measures (DPM) may be the most effective approach to control the rapid spread of the disease and to sustain healthcare systems. From an epidemiological perspective, person-to-person contact through human behaviours can be a fundamental means for virus transmission [2]. To slow down the transmission of the virus, the large-scale adoption of DPM to minimise direct interpersonal contact is essential [3]. To prevent the spread of COVID-19, the World Health Organization has recommended the avoidance of interpersonal contact between infected and non-infected individuals, the implementation of early detection and case isolation, and the adoption of general individual and collective hygiene measures [4]. In practice, community-level DPM include lockdowns, curfews, the isolation of infected populations, bans on social gatherings, and mandatory home quarantine, whilst individual-level DPM may involve the promotion of wearing face masks in public areas, frequent hand washing, and social distancing.

The effectiveness of DPM to control the spread of COVID-19 depends heavily on the public’s adherence. Unfortunately, the public’s adherence to different DPM varies to a great extent across countries. For example, the adherence rates range from 22% (Uganda) to 96% (Macau) for wearing face masks [5,6]; from 79% (Macau) to 98% (Brazil) for frequent hand washing [6,7]; and from 58% (the Democratic Republic of Congo) to 85% (Japan) for social distancing [8,9]. In Hong Kong, one of the first places affected by COVID-19, the government recommended several major DPM at individual levels, which include the wearing of face masks in public areas, the avoidance of social gatherings, social distancing, the avoidance of touching eyes, nose, and mouth, hand washing, the use of alcohol-based hand rubs, and voluntary testing for COVID-19. Despite a scarcity in empirical findings on the adherence to DPM in Hong Kong, a strong adherence among its citizens could be expected given the lessons learnt from the Severe Acute Respiratory Syndrome (SARS) pandemic [10]. Indeed, a recent observational study in Hong Kong reported a 75–99% rate of wearing face masks and 61–86% rate of avoidance of crowded places, and the strong adherence to DPM may contribute to the relatively low morbidity and mortality at the beginning of the COVID-19 pandemic in the city [3]. However, high-profile incidents of non-adherence appeared in later stages of the pandemic and were believed to result in various waves of COVID-19 infection. The non-adherence highlights the challenges for public health professionals and other stakeholders to understand the mechanisms behind one’s motivation to adhere to guidelines and to determine how best to encourage the public to adopt and maintain the fullest extent of DPM during the pandemic.

Cognitive health behaviour theories indicate that individuals’ beliefs and attitudes can account for their adoption of health-promoting behaviours [11]. The Health Belief Model (HBM) [12], one of the most commonly used frameworks to explain health behaviours, may be a useful tool to guide the promotion of COVID-19 DPM. The main elements of the HBM, which focus on one’s beliefs on health conditions and the related health behaviours, include: (i) perceived susceptibility, which is the perceived risk of contracting the disease; (ii) perceived benefits, the positive consequences of adopting the needed behaviours; (iii) perceived barriers, the tangible and psychological costs of adopting the behaviours; (iv) self-efficacy, the perceived ability to perform the behaviours; and (v) cues to action, the external stimuli triggering the adoption of the behaviours.

Understanding the factors affecting the public’s adherence to DPM is the key to the successful promotion of behaviours that help control the spread of COVID-19. Using a probability community sample of Chinese adults (>18 years of age), this study aimed to explore the level of adherence to various DPM in the Hong Kong population approximately one year after COVID-19 first emerged in the city, and to explain and predict the adherence to DPM by identifying cognitive factors based on the HBM. In this study, we extended the HBM by including the perceived acceptability of DPM implemented by governments (e.g., regulations on wearing masks). Additionally, based on the findings in past research [13,14,15], we added several psychosocial and cognitive factors in the prediction model for adherence to DPM, namely perceived disruptions in daily life related to COVID-19, knowledge about the disease, and trust in authorities (e.g., governments and healthcare systems), and examined their effects. It was hypothesised that, when confounding factors such as demographic characteristics were adjusted for, the HBM factors as well as the psychosocial and cognitive factors included would significantly predict one’s adherence to DPM.

## 2. Methods

### 2.1. Design and Sample

We conducted a cross-sectional telephone survey from December 2020 to January 2021 among a probability community sample of adults aged 18 years or above in Hong Kong. Telephone numbers were sampled using a multi-stage procedure. We first drew telephone numbers randomly from residential telephone and mobile directories. Using them as “seed” numbers, we then generated another set of telephone numbers with the “plus/minus one/two” method to capture unlisted numbers and created the final set of numbers by filtering duplicated ones. A team of trained interviewers was responsible for contacting eligible participants and conducting structured interviews via telephone calls under close supervision of our research team. All adults aged 18 years or above who were residents of Hong Kong and able to understand Cantonese were invited to participate.

In this study, a total of 6000 telephone numbers were sampled with the multi-stage procedure. Of these 6000 numbers, 3216 were not valid (e.g., fax line numbers, numbers not in use, etc.). We contacted the remaining 2784 numbers and successfully completed 1255 interviews (response rate = 45.1%). The response rate of 45.1% was comparable to those of population-based telephone survey studies in Hong Kong, which ranges from 41% to 47% [16,17]. Excluded cases included refusal (19.4%), non-contact (34.8%), and language barriers (0.8%).

Participants gave oral consent over the phone and responded to a questionnaire which elicited their demographic background, adherence to DPM, HBM factors, awareness and knowledge of COVID-19, perceived disruptions in daily life related to COVID-19, and trust in authority via phone calls.

### 2.2. Measures

The study outcome was the level of adherence to six DPM recommended by the Hong Kong government, which included wearing face masks properly in public areas; avoiding touching one’s eyes, nose, and mouth; performing hand hygiene practices with soap; using alcohol-based hand rubs before eating food and after using the toilet; performing social distancing and avoiding group gatherings; and participating in voluntary testing for COVID-19. Participants were asked to report their level of adherence to each DPM on a 5-point scale, from 1 (never or almost never) to 5 (always). The item scores were averaged to give an overall adherence to COVID-19 DPM; the higher the score, the greater adherence to DPM.

In this study, we extended the HBM to cover the following aspects: (i) perceived susceptibility to COVID-19, which was assessed by a single item “I am likely to be infected with COVID-19.”; (ii) perceived benefits of DPM, which was measured with three items (e.g., “Adopting DPM prevents COVID-19.”); (iii) perceived barriers, which was measured with six items (e.g., “It is difficult for me to obtain protective equipment such as face masks and hand rub.”); (iv) self-efficacy, which was assessed with the 10-item Generalized Self-Efficacy Scale (e.g., “I can solve most problems if I invest the necessary effort.”) [18]; (v) cues to action, which was assessed with seven items (e.g., “My doctor or nurse thinks that I should adopt DM.”); and (vi) perceived acceptability of DPM, which was measured with six items (e.g., “Compulsory testing for COVID-19 among suspected cases is reasonable.”). All items were rated on a 5-point scale, from 1 (strong disagree) to 5 (strongly agree). Mean scores of scales with more than one item were computed, and higher scores indicated higher levels of agreement.

In addition to the HBM factors, we assessed three other psychosocial and cognitive variables among the participants. Disruptions to daily life related to COVID-19 were assessed with a list of nine items, capturing the degree of disturbance of various aspects of living routines caused by or related to COVID-19 (e.g., disturbance to one’s social life, family interactions, etc.). On the other hand, trust in authority was assessed with two items (e.g., “I trust that the government will be able to control COVID-19.”). Similar to the HBM factors, all items were rated on a 5-point scale from 1 (strongly disagree) to 5 (strongly agree). Higher item scores and mean scores reflected a higher degree of agreement.

Participants’ knowledge of issues related to COVID-19 was also assessed using the 12-item COVID-19 Knowledge Scale [19]. The scale consisted of 12 statements about the symptoms, transmission route, and prevention methods of COVID-19. Sample statements included “Not everyone with COVID-19 has a severe case of the disease” and “The main clinical symptoms of COVID-19 are fever, fatigue, dry cough, and myalgia”. Participants were asked to decide whether the statement was correct or not, by choosing among “true”, “false”, and “don’t know”. Total scores ranged from 0 to 12, with higher scores indicating better knowledge of COVID-19 and its related issues.

Demographic characteristics including gender, age, education attainment, and economic activity status of the participants were recorded and controlled as confounding variables in the data analysis.

### 2.3. Statistical Analysis

We conducted all analyses in IBM SPSS 24.0 (IBM Corp, Armonk: NY, USA). Descriptive data on the demographic characteristics and the adherence to DPM were computed and summarised. Means, standard deviations, and Cronbach’s alphas were calculated for all measures. To ensure the representativeness of the final sample, the raw data collected were rim-weighted by the latest sex-age distribution and education attainment distribution in the Hong Kong population. Associations among adherence to DPM, the HBM factors, disruptions in daily life, trust in authority, and knowledge were explored with Pearson’s *r* for bivariate correlations. To examine the effect of each factor on the adherence to DPM, multinomial hierarchical regressions were performed, and the main effects of the HBM factors on adherence to DPM were estimated with the adjustment of potential confounding factors such as gender and age. In the hierarchical regression analysis, we included demographic factors (i.e., gender, age, education attainment, and economic activity status) in step 1, the six HBM factors in step 2, and other psychosocial and cognitive variables (including knowledge, disruptions in daily life, and trust in authority) in step 3. All analyses were based on two-sided *p*-values. *p*-values smaller than 0.05 were considered as statistically significant.

## 3. Results

### 3.1. Demographic Characteristics

Table 1 presents the demographic characteristics of the sample. The majority of the sample (*N* = 1255) were women (53.0%). About 41.3% of the sample were senior adults aged 55 years or above, 35.0% were between 35 and 54 years, and 23.7% were young adults aged between 18 and 34 years. Most participants had received a high-school education (45.6%) or above (23.0%) and were economically active during the study period (61.7%). We conducted gender comparisons on the demographic characteristics and found significant differences only in the economic activity status, where a greater proportion of men was economically active than women (70.1% versus 54.2%, *p* < 0.001).

### 3.2. Adherence to DPM

Participants’ responses to their level of adherence to each DPM were weighted and are summarised in Table 2 and Table 3. Overall, participants’ self-reported adherence ratings were high, with an average of 4.12 (SD = 0.60) which was close to the top of the five-point scale. Taking reference from a previous study in Macau,^5^ the level of adherence was considered as strong with a practice frequency of “often” or “always”. In this study, wearing face masks in public areas was the DPM with highest level of adherence (94.4%); followed by the avoidance of touching eyes, nose, and mouth (88.4%); performance of hand hygiene practices (82.1%); and the use of alcohol-based hand rubs (81.5%). Almost three-quarters of participants showed strong adherence to social distancing (74.6%). Among all these, voluntary testing for the disease was the least adhered DPM, and less than half of the participants adhered to it (39.7%).

### 3.3. Factors Associated with Adherence to DPM

The mean scores and standard deviations of the HBM factors as well as other psychosocial and cognitive factors are presented in Table 3. Concerning the knowledge about COVID-19, the mean score was 9.32 out of a 12-point scale (SD = 2.34), which suggested an overall correct rate of 77.7% among the participants.

The findings of the Pearson’s correlation analyses indicated moderate to large correlations of some HBM factors with adherence to DPM. Specifically, perceived benefits (*r* = 0.46, *p* < 0.01), cues to action (*r* = 0.47, *p* < 0.01), and perceived acceptability of DPM (*r* = 0.54, *p* < 0.01) were positively associated with adherence, whilst perceived barriers were inversely associated (*r* = −0.28, *p* < 0.01). Among other psychosocial and cognitive variables, knowledge (*r* = 0.28, *p* < 0.01), disruptions in daily life (*r* = 0.15, *p* < 0.01), and trust in authority (*r* = 0.25, *p* < 0.01) were positively correlated with the level of adherence with a small to moderate effect size.

Table 4 summarises the findings of the hierarchical regression analysis. The results showed that the HBM factors, together with demographic characteristics, predicted 36.6% of the variance of participants’ levels of adherence to DPM (*F*_(10, 1244)_ = 71.88, *p* < 0.001). After adjustment for the demographic characteristics, perceived benefits (B = 0.12, *p* < 0.001), self-efficacy (B = 0.07, *p* = 0.004), cues to action (B = 0.23, *p* < 0.001), and perceived acceptability of DPM (B = 0.29, *p* < 0.001) were found to be significant predictors of adherence to DPM.

When knowledge, disruptions in daily life, and trust in authority were added in the regression analysis, the final model accounted for 38.2% of the variance of adherence to DPM (*F*_(13, 1241)_ = 58.99, *p* < 0.001). HBM factors including perceived benefits (B = 0.10, *p* < 0.001), self-efficacy (B = 0.07, *p* = 0.002), cues to action (B = 0.17, *p* < 0.001), and perceived acceptability of DPM (B = 0.30, *p* < 0.001) remained as significant predictors of adherence to DPM. In the final model, the negative association between perceived barriers and adherence became statistically significant (B = −0.05, *p* = 0.005). After adjustment for demographic characteristics and HBM factors, disruption in daily life (B = 0.09, *p* < 0.001) was significantly associated with participants’ adherence to DPM. The findings of the hierarchical regression in this study, however, did not reveal any significant association between perceived susceptibility, knowledge and trust in authority, with adherence to DPM among participants.

## 4. Discussion

In the absence of a definitive pharmaceutical approach of prevention or treatment of COVID-19, self-imposed DPM in order to control the rapid spread of the disease may serve as an essential means to drastically reduce the rate of morbidity and mortality within a population. This study used a probability community sample of Chinese adults to examine their adherence to various DPM recommended the Hong Kong government. The findings demonstrated generally strong adherence to wearing of proper face masks, avoidance of touching eyes, nose, and mouth, performance of hand hygiene practices, use of alcohol-based hand rubs, and abiding by social distancing, which were often or always complied by 75–94% of participants. The adherence rates were comparable to those revealed in past studies worldwide, which found that 96% wore face masks in public areas (96%) [6]; 85% practised social distancing [8]; 79–98% performed hand hygiene practices and washed hands frequently [7,9,20]; and 79–95% used hand sanitizers [21,22]. In this study, voluntary testing for COVID-19 was the least adhered DPM. The adherence rate was 39.7%, which was comparable to the testing rate in other countries such as New Zealand (42%), Japan (9%), and South Korea (17%) [23], where mass asymptomatic testing for COVID-19 has not yet been implemented.

In congruence with some Asian research [5], we observed cultural differences in the adherence to some DPM when compared our findings with those in the European, U.S., and African studies. For example, there was a relatively stronger adherence to wearing face masks in the Chinese population than that among their Western counterparts. In a U.S. study [24], wearing face masks was the least adhered to DPM and the overall adherence level was “unlikely”; in some African studies, this specific DPM was poorly complied among citizens (6–46%) [8,25]. Some researchers believed that shortages of face masks and other equipment in some countries could be the major underlying reason for the poor adherence [8], yet, others suggested possibilities of differences in common health beliefs and practices across populations. In this sense, Asian populations, especially the Japanese population in which wearing face masks is common during flu seasons and the Hong Kong population who had tragic experience during the SARS outbreak, may be less hesitant than other populations to wear proper face masks in public areas, leading to an overall stronger adherence. Another potential cultural difference can be observed in the adherence to social distancing. Our findings, together with those revealed in a Macau study [6], showed that the Chinese population were less adherent to social distancing or the avoidance of gatherings (43–75%), whilst those in a U.S. study demonstrated a relatively strong adherence that individuals were in general “very likely” to avoid crowded areas [24]. To explain the gap between the adherence to social distancing and that to other hygiene behaviours, some suggested that these two types of DPM might be driven by different mechanisms. For example, positive attitudes toward authorities were associated with adherence to social distancing measures but not hygiene behaviours [26]. Since adherence to social distancing may involve more restrictive changes in behaviours, it could easily lead to non-adherence when individuals’ trust and belief that the authorities are effective and fair in implementing the relevant regulations are insufficient [23]. Indeed, our findings showed that trust in authority was positively correlated to the overall adherence. Future research may explore whether there are differences in the underlying mechanisms accounting for the varying adherence levels of DPM.

This study shed lights on the underlying mechanisms of COVID-19 DPM by demonstrating that HBM factors play significant roles in altering one’s adherence level. Overall, the HBM factors could explain an extra 34% of variance of DPM adherence on top of demographic characteristics, reflecting the robust effects of the psychosocial and cognitive factors in our extended HBM. Consistent with past findings [20,27,28,29], this study revealed significant effects of the perceived benefits, perceived barriers, and self-efficacy. According to the original HBM theory, the product of the net effect of perceived benefits and barriers could convert one’s intention to adopt health-promoting behaviours into action [30]. In this study, individuals who believed that DPM could be beneficial to oneself as well as to others, those who did not find obstacles in adopting DPM (e.g., difficulties in getting masks and hand rub), and those who had high level of self-efficacy in general reported stronger adherence to DPM. Based on our findings, future promotion campaigns may incorporate messages to highlight the importance of DPM in providing benefits to oneself as well as the community as a whole, so as to convince the public that DPM could lead to a valued outcome. Furthermore, to boost one’s intention to act and adherence, the authorities should identify specific barriers perceived by the public and provide relevant assistance to help individuals adhere to DPM on a timely basis. Strategies to maximise perceived benefits and minimise perceived barriers might then further increase one’s self-efficacy specific to COVID-19 DPM, which in turn bridge the gap between intention and action and strengthen the public’s adherence.

In this study, the perceived acceptability of DPM had the greatest positive effect on DPM adherence: Individuals who showed greater acceptability to the regulations related to COVID-19 prevention implemented by the government were more willing to adhere to DPM. This result implied the importance of well-organised communications by the government and authorities to explain the reasons behind each COVID-19-related regulation, so as to enhance the public’s belief on the effectiveness and fairness of the government regulations. Indeed, studies have demonstrated the effectiveness of government communications that were guided by public’s interests and feelings in promoting individuals’ perceived usefulness and adherence of DPM [31].

It is noteworthy that in our regression model, disruption was the only significant associated factor apart from the HBM factors. Since adherence to DPM involves great behavioural changes in daily life, a greater disruption in daily life could be expected among individuals who better adhered to the measures. Surprisingly, knowledge about COVID-19, which has been shown to be a robust predictor of adherence to DPM in other studies [32,33], did not contribute significantly to DPM adherence in this study. This reflects that individuals in Hong Kong were willing to adopt and adhere to DPM despite the fact that they did not have accurate knowledge on COVID-19. This might be partly explained by the tragic experience of the SARS pandemic during which Hong Kong bore a large proportion of the world’s morbidity and mortality burden [34]. Since then, citizens have learnt to prevent virus transmission by various DPM including wearing face masks and hand washing [10,35]. These experiences might simply drive individuals to adopt DPM without having a deep understanding of COVID-19, leading to a strong adherence level independent of accurate knowledge about the disease.

In this study, several limitations should be noted. First, this study might not be able to provide sufficient evidence for casual relationships between adherence to DPM and other variables. Whether the HBM factors led to the adherence of DPM or vice versa was unknown. The use of self-reported measures to collect retrospective data, which might involve self-reporting and recall biases, was another limitation. We could not deny the possibility of socially desirable responses given by the participants. Due to the constraints by the COVID-19 pandemic, this study used telephone surveys to collect data. The response rate of this study (45%) might not be comparable to that of studies using other means of data collection (e.g., face-to-face household interviews or online surveys). Since about two-thirds of the non-response cases in this study were unable to be reached, the demographic backgrounds of eligible individuals who did not participate are missing. The fact that we could not compare the data between participants and “non-participants” raises a possibility of bias. For example, adherence to DPM, the outcome of this study and other independent variables might affect the participation. This might induce a potential collider bias that could distort the associations between variables in the sample. Although we weighted the data by the latest sex-age and education attainment distribution, there could still be sampling bias that limited the generalisability of our findings. Finally, as one of the preliminary efforts in the world to understand individuals’ adherence to COVID-19 DPM, our study focused on the population in Hong Kong only. Future research may be conducted in other regions for international comparisons to extend current knowledge (e.g., other cities in Asia, Europe, North America, etc.).

## 5. Conclusions

Our past experiences in the SARS pandemic and other epidemics suggest that simple behavioural changes at the individual level could serve as a low-cost but effective way for controlling pandemics [36]. While COVID-19 vaccination programmes have begun in some countries, the long-term effectiveness of the vaccines to control the spread is still unknown. At this stage, strong adherence to DPM among the public continues to serve as the utmost important means to combat the spread of the disease. This study showed a strong adherence to DPM among the Hong Kong population and demonstrated that such adherence could be amenable to improvement. The findings shed new lights on promoting DPM and other health behaviours at both individual and community levels, by highlighting the importance of perceived benefits, perceived barriers, self-efficacy, cues to action, and perceived acceptability in promoting adherence to the DPM recommended by the authorities. Based on the results, future campaigns for the promotion of COVID-19 DPM should involve (i) effective communications by the authority to thoroughly explain to the public the reasons behind and the benefits of the DPM so as to enhance one’s acceptance level; (ii) efforts to minimise the potential barriers of the DPM; and (iii) more cues for the public to convert their intention into action.

## Figures and Tables

**Table 1 ijerph-18-12403-t001:** Demographic characteristics among study participants.

		Total (*N* = 1255)				Gender Comparison							
						Male (*n* = 598)				Female (*n* = 657)			*p* -Value
	*n*	%	Weighted n ^a^ (*N* = 6,134,040)	Weighted %	*n*	%	Weighted n ^a^ (*n* = 2,881,760)	Weighted %	*n*	%	Weighted n ^a^ (*n* = 3,252,280)	Weighted %	
Gender													
Female	657	52.4	3,252,280	53.0									
Male	598	47.6	2,881,760	47.0									
Age (years)													0.72
18–34	294	23.4	1,452,440	23.7	144	24.1	716,160	24.9	150	22.8	737,280	22.7	
35–54	442	35.2	2,147,800	35.0	204	34.1	960,500	33.3	238	36.2	1,187,300	36.5	
55 or above	519	41.4	2,532,800	41.3	250	41.8	1,205,100	41.8	269	40.9	1,327,700	40.8	
Education attainment													0.26
Junior high school or below	392	31.2	1,927,535	31.4	171	28.6	816,714	28.3	221	33.6	1,110,821	34.2	
High school	575	45.8	2,795,408	45.6	282	47.2	1,357,175	47.1	293	44.6	1,438,234	44.2	
Tertiary or above	288	11.3	1,411,096	23.0	145	24.2	707,871	24.6	143	21.8	703,225	21.6	
Economic activity status													<0.001
Active	775	61.8	3,782,101	61.7	417	69.7	2,020,568	70.1	358	55.5	1,761,532	54.2	
Inactive	480	38.2	2,351,939	38.3	181	30.3	861,192	29.9	299	44.5	1,490,748	45.8	

^a^ Data were weighted by the latest sex-age distribution and education attainment distribution in the Hong Kong population.

**Table 2 ijerph-18-12403-t002:** Unweighted and weighted rates of adherence to DPM.

	Unweighted % (*N* = 1255)					Weighted % ^a^ (*N* = 3,252,280)				
	Never/Almost Never	Occasionally	Sometimes	Often	Always	Never/Almost Never	Occasionally	Sometimes	Often	Always
Wearing face masks properly in public areas	0.1	2.1	3.5	12.7	81.6	0.1	2.1	3.5	12.8	81.6
Avoiding touching one’s eyes, nose, and mouth	0.6	2.1	8.8	24.1	64.4	0.6	2.1	8.8	24.1	64.3
Performing hand hygiene practices with soap	0.2	3.7	13.9	37.2	44.9	0.2	3.7	13.9	37.1	45.0
Using alcohol-based hand rubs before eating food and after using the toilet	0.5	3.7	14.3	33.2	48.3	0.5	3.8	14.3	33.2	48.3
Performing social distancing and avoiding group gathering	1.1	5.9	18.2	34.2	40.6	1.1	5.9	18.3	34.1	40.5
Participating in voluntary testing of COVID-19	26.6	14.6	18.9	20.2	19.6	26.6	14.7	18.9	20.2	19.5

^a^ Data were weighted by the latest sex-age distribution and education attainment distribution in the Hong Kong population; COVID-19 = Coronavirus disease

**Table 3 ijerph-18-12403-t003:** Weighted mean scores, SD, and Cronbach’s alpha of the adherence to DPM, HBM factors, awareness of COVID-19, knowledge, perceived disruptions in daily life, and trust in authority.

	Range of Scale Score	Weighted Mean Score ^a^ (*N* = 6,134,040)	SD	Cronbach’s Alpha
Adherence to DPM	1–5	4.12	0.60	0.68
HBM factors				
Perceived susceptibility to COVID-19	1–5	2.94	0.91	-
Perceived benefits of DPM	1–5	4.23	0.67	0.78
Perceived barriers	1–5	2.54	1.00	0.88
Self-efficacy	1–5	3.42	0.73	0.92
Cues to action	1–5	4.07	0.58	0.83
Perceived acceptability of DPM	1–5	4.40	0.66	0.85
Knowledge of COVID-19				
Total score	0–12	9.32	2.34	0.72
Symptoms	0–4	2.82	1.02	0.38
Mode of transmission	0–3	2.21	0.80	0.19
Prevention and control	0–5	4.29	1.25	0.77
Disruptions in daily life related to COVID-19	1–5	3.37	0.89	0.88
Trust in authority	1–5	3.72	0.78	0.68

^a^ Data were weighted by the latest sex-age distribution and education attainment distribution in the Hong Kong population. SD = Standard deviation. DPM = Disease prevention measures. HBM = Health belief model. COVID-19 = Coronavirus disease.

**Table 4 ijerph-18-12403-t004:** Hierarchical multiple regression model predicting adherence to DPM among study participants (*N* = 1255).

	Step 1			Step 2			Step 3		
	B	SE(B)	*p*-Value	B	SE(B)	*p*-Value	B	SE(B)	*p*-Value
Demographic characteristics									
Gender ^a^	0.11	0.09	0.002	0.03	0.03	0.300	0.04	0.03	0.17
Age	0.00	0.11	0.002	0.00	0.00	0.931	0.00	0.00	0.46
Education attainment	0.06	0.08	0.038	0.01	0.02	0.813	0.00	0.02	0.93
Economic activity status ^a^	−0.16	−0.13	<0.001	−0.11	0.03	<0.001	−0.11	0.03	0.001
HBM variables									
Perceived susceptibility of COVID-19				−0.02	0.04	0.660	−0.04	0.04	0.30
Perceived benefits of DPM				0.12	0.03	<0.001	0.10	0.03	<0.001
Perceived barriers				−0.03	0.02	0.058	−0.05	0.02	0.005
Self-efficacy				0.07	0.02	0.004	0.07	0.02	0.002
Cues to action				0.23	0.03	<0.001	0.17	0.03	<0.001
Perceived acceptability of DPM				0.29	0.03	<0.001	0.30	0.03	<0.001
Knowledge on COVID-19 (total score)							0.01	0.01	0.11
Disruptions in daily life related to COVID-19							0.09	0.02	<0.001
Trust in authority							0.03	0.02	0.10

Model statistics									
R-squared	0.024			0.366			0.382		
F	7.61			71.88			58.99		
*p*-value	<0.001			<0.001			<0.001		

DPM = Disease prevention measures. HBM = Health belief model. COVID-19 = Coronavirus disease. ^a^ Referent group = Gender (male); economic activity status (economically active). B = unstandardized regression coefficient. SE(B) = standard errors for regression coefficients. F = F changed.

## Data Availability

Materials and anonymous data are available from the authors by request.

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
