# Peer review of "Predicting Public Adherence to COVID-19 Preventive Measures: A Cross-Sectional Study in Hong Kong"

_ijerph, 2021, doi:10.3390/ijerph182312403_

Round 1
Reviewer 1 Report
- In the Discussion session, I suggest the authors talk more about the limitations of this study and their impacts on results, especially the impacts of potential selection bias. The authors mentioned in the Methods section that the response rate of the telephone survey is 45%. This can raise concerns from two aspects: 1) The estimated % DPM adherence among the participants may not reflect the general population; regarding this issue, I appreciate that the authors provided weighted estimates based on demographic distributions - as long as the participants are representative of the population within each demographic category (i.e., female participants are representative of the overall female population), this should be fine. 2) My concern is more related to the estimated associations between DPM and various other factors (demographic, HBM, and psychosocial/cognitive factors). If both the DPM adherence (the outcome) and other factors under study (the exposure) both affect the participation (i.e., collider bias), then a spurious association can be induced between the DPM adherence and those other factors.
- In the Discussion section, the last paragraph, the authors wrote "Like all other cross-sectional research, this study could not provide evidence for causal relationships between ..." - this statement regarding cross-section research is unfair because cross-sectional study can be used to address causal questions. I suggest the authors focus the discussion on the limitations of this specific study, instead of generalizing the limitation to all other cross-section research.
- Please clarify the time period when the telephone survey was conducted. The Abstract said "December 2019 - January 2021", while the main text said "December 2020 - January 2021".
- In Table 3, the authors present scores for various factors. Because they are on different scales (some scale from 1-5, while others from 1-12, for example), I wonder if the authors can add another column to show the corresponding scales - this may help with an easier read.
Reviewer 2 Report
There are some comments that author should be considered.
- "In this study, a total of 6,000 telephone numbers were sampled with the multi-stage procedure......We contacted the remaining 2,784 numbers and successfully completed 1,255 interviews (response rate = 45.1%)." The final sample size is very different from the design sample. Whether there are differences between desgin sample and collected sample, such as gender, educaiton, age.
- "To ensure the representativeness of the final sample, the raw data collected were rim-weighted accord-ing to the latest sex-age distribution and education attainment distribution in the Hong Kong population." How this method fit on the final sample? Could you provide more information on the original sample. The weighted usally according to the design not according to the "atest sex-age distribution and education attainment distribution in the Hong Kong population", unless the surey design fits it.
- "hierarchical regression" use weighted or unweighted? Because in the tables author presented weighted number.
- "DPM" please added some information on it
Reviewer 3 Report
Title: Predicting Public Adherence to COVID-19 Preventive Measures: A Cross-sectional Study in Hong Kong.
This study examined individuals’ adherence to various DPM, and identified facilitators and barriers to adherence in a community sample in Hong Kong. The study methodology relied on using telephone surveys were conducted during December 2019 and January 2021. Participants provided responses on their adherence to DPM as well as other psychosocial and cognitive factors via the phone. The presentation of the results was good and there was a clear link between the introduction, results and discussion of findings. The implications of the findings can prove useful for potential future pandemics in understanding people’s behaviour and motivations for adherence to certain DPMs. Limitations were sufficiently acknowledged as it was unknown the extent to which bias affected respondents answers to the researchers.
A follow-up study in other regions (e.g. Europe, Africa, America) would be interesting for comparisons.
There were no obvious or apparent spelling or grammar mistakes.
Round 2
Reviewer 2 Report
The author had addressed all the comments I previously mentioned.